# Dedicated Algorithm for Unobtrusive Fetal Heart Rate Monitoring Using Multiple Dry Electrodes

**DOI:** 10.3390/s21134298

**Published:** 2021-06-23

**Authors:** Alessandra Galli, Elisabetta Peri, Yijing Zhang, Rik Vullings, Myrthe van der Ven, Giada Giorgi, Sotir Ouzounov, Pieter J. A. Harpe, Massimo Mischi

**Affiliations:** 1Department of Information Engineering, University of Padova, I-35131 Padova, Italy; galliale@dei.unipd.it (A.G.); giada@dei.unipd.it (G.G.); 2Department of Electrical Engineering, Eindhoven University of Technology, 5600 MB Eindhoven, The Netherlands; e.peri@tue.nl (E.P.); yijing.zhang@tue.nl (Y.Z.); r.vullings@tue.nl (R.V.); m.v.d.ven1@tue.nl (M.v.d.V.); p.j.a.harpe@tue.nl (P.J.A.H.); 3Philips Research, 5656 AE Eindhoven, The Netherlands; sotir.ouzounov@philips.com

**Keywords:** fetal heart rate monitoring, artifacts removal, blind source separation, dry electrodes, textile electrodes, multi-channel measurements

## Abstract

Multi-channel measurements from the maternal abdomen acquired by means of dry electrodes can be employed to promote long-term monitoring of fetal heart rate (fHR). The signals acquired with this type of electrode have a lower signal-to-noise ratio and different artifacts compared to signals acquired with conventional wet electrodes. Therefore, starting from the benchmark algorithm with the best performance for fHR estimation proposed by Varanini et al., we propose a new method specifically designed to remove artifacts typical of dry-electrode recordings. To test the algorithm, experimental textile electrodes were employed that produce artifacts typical of dry and capacitive electrodes. The proposed solution is based on a hybrid (hardware and software) pre-processing step designed specifically to remove the disturbing component typical of signals acquired with these electrodes (triboelectricity artifacts and amplitude modulations). The following main processing steps consist of the removal of the maternal ECG by blind source separation, the enhancement of the fetal ECG and identification of the fetal QRS complexes. Main processing is designed to be robust to the high-amplitude motion artifacts that corrupt the acquisition. The obtained denoising system was compared with the benchmark algorithm both on semi-simulated and on real data. The performance, quantified by means of sensitivity, F1-score and root-mean-square error metrics, outperforms the performance obtained with the original method available in the literature. This result proves that the design of a dedicated processing system based on the signal characteristics is necessary for reliable and accurate estimation of the fHR using dry, textile electrodes.

## 1. Introduction

The prevalence of high-risk pregnancies is increasing because of the progressively higher age at which women become pregnant. Nowadays, about 20% of pregnancies are complicated [1]. Early detection and diagnosis of the complications are fundamental to guarantee timely medical intervention, but the existing monitoring techniques preclude continuous monitoring of fetal well-being.

One of the most important parameters to evaluate fetal health is the fetal heart rate (fHR), which is commonly measured with the cardiotocograph (CTG). CTG measurements are based on Doppler ultrasound, which is limited by the high sensitivity to the fetal position. Indeed, due to fetal movement or displacement of the transducer, the relative fetal heart location with respect to the ultrasound transducer can change, leading to frequent periods of signal loss [2]. Good results are obtained only if the transducer is frequently repositioned by specialized medical staff, making long-term monitoring in a domestic environment impractical.

Recently, to overcome the CTG limitations, devices based on the acquisition of electrophysiological signals have been proposed [3]. This type of signal can be acquired with simple and less invasive instrumentation, and the quality of the signal is not strongly correlated to the position of the fetal heart. The electrophysiological signal is acquired on the maternal abdominal surface through electrodes and is a mixture of several components. This application traditionally makes use of adhesive wet electrodes, such as Ag/AgCl electrodes. Although contact electrodes provide good signal quality, they are uncomfortable and can cause adverse reactions, especially if the electrodes are worn for a long time. In addition, the quality of the acquired signal decreases with gel dehydration due to prolonged use. For these reasons, adhesive gel electrodes are not suitable for long-term monitoring [4].

Dry or capacitive sensing seems promising in ECG acquisition [5]: the electrodes can be integrated into wearable materials making the acquisition comfortable and suitable for long-term monitoring without causing skin irritation. On the other hand, the lack of gel between the skin and the electrode increases the contact impedance, which results in lower signal amplitude. Furthermore, due to the lack of contact, these electrodes are subject to several artifacts [6], which make fHR extraction even more complex than using traditional instruments and sensors [7].

Several works were presented in the literature for the detection of the fetal QRS (fQRS) complexes to provide a reliable and robust estimation of fHR by means of the electrophysiological signals acquired with standard wet electrodes. Approaches presented in the literature can be divided into two groups. The first proposes to improve the detection of the fetal electrocardiographic (fECG) by maternal ECG (mECG) removal, through asynchronous averaging [8,9], by means of adaptive filtering [10] and by using the subspace approximation of the full ECG cardiac complex [11,12]. In the approaches of the second group, the components are separated by blind source separation (BSS) methods. The employed BSS techniques are Principal Component Analysis (PCA) [13], Singular Value Decomposition (SVD) [14] or Independent Component Analysis (ICA) [15]. In addition, the method proposed in [16] is based on the combination of these two approaches.

In this work, we employ dry, textile electrophysiological sensors, which have the advantage of comfortably acquiring the signals and are hence fully unobtrusive. These electrodes produce artifacts that are different and larger than those produced by wet electrodes. To the best of our knowledge, there are no methods in the literature that are specifically designed for processing signals acquired with dry electrodes. The algorithms mentioned above have very poor fHR estimation performance because they are based on signals acquired with wet electrodes. In this work, we propose to fill this gap by an algorithm for fHR estimation that is based and designed on signals acquired by dry electrodes. In detail, we modified the algorithm proposed by Varanini et al. [16], referred to as VA, and adapted it to the characteristics of the acquisitions made with dry electrodes. Our novel contribution, discussed in Section 2.3, consists of the introduction of a hybrid pre-processing stage, consisting of a combined hardware and software solution, designed specifically to remove the disturbing component typical of acquisitions by dry electrodes (triboelectricity artifacts and amplitude modulations). In addition, we improved the main processing, i.e., the removal of the mECG by BSS, the enhancement of the fECG and the identification of the fQRS complexes, in order to make the denoising robust to the typical motion artifacts that affect this type of acquisition.

The obtained denoising system was tested both on semi-simulated and real data, which are described in Section 2.2. The metrics employed to quantify the performance are Sensitivity, F1-score and root-mean-square error (RMSE), described in Section 2.4 and compared by statistical analysis (Section 2.5) with the performance of the benchmark algorithm from Varanini et al. [16]. Results are reported in Section 3 and discussed in Section 4.

## 2. Materials and Methods

### 2.1. Acquisition System

Design and realization of the acquisition system, shown in Figure 1a, was a part of a larger project involving the collaboration between Eindhoven University of Technology (TU/e), Philips Research in Eindhoven and Máxima Medical Center (MMC) in Veldhoven.

As shown in Figure 1b, textile electrodes made of conductive fabric are connected to the prototype amplifier through 1.5-m wires. Even though the adopted fabric is conductive, the high resistance still causes charge accumulation similar to capacitive electrodes, producing artifacts due to triboelectricity. In this study, the textile electrodes were positioned directly on the skin and held in place by means of elastic belts. In particular, four electrodes were placed on the belly, symmetrically with respect to the navel, and a reference electrode, i.e., the ground, was placed on the hip. The electrode configuration scheme is shown in Figure 1c. This configuration was chosen because it maximizes the SNR [17]. The sampling rate of the instrument was $F_{s}=1000$ Hz and the data were acquired with a resolution of 14 bits. It is a wearable system supplied by batteries.

### 2.2. Dataset

The proposed algorithm was tested with two datasets consisting of semi-simulated (DS-SS) and real (DS-R) acquisitions. In the DS-SS dataset, the acquisitions were performed on the abdomen of non-pregnant women, and the fECG was simulated and superimposed thoroughly in order to obtain reference values to assess the performance of the algorithm. According to physiology, since the mECG and the fECG components are generated by different sources, the best model to merge the real part with the simulated component was the additive model. Simulated fECG was obtained through an online simulator [18]. For each acquisition, a different fECG was generated and its amplitude was randomly selected in the range of 10–20 μVpp. According to the amplitude of the fECG, the resulting SNR of DS-SS dataset was −48±6 dB. DS-SS contained 27 semi-simulated subjects, and the length of each multi-channel measurement was one minute.

The (DS-R) dataset consisted of multi-channel measurements performed on pregnant women. The acquisitions were carried out at the labor ward of Máxima MC Veldhoven, Netherlands. The inclusion criteria for women’s recruitment were the following:Gestational age (GA) >36 weeks;Fetus in cephalic position;Body Mass Index (BMI) <30 (before pregnancy);Singleton pregnancy.

On the other hand, the exclusion criteria for the pilot study were:Age <18 years;Pregnancy with maternal or fetal complications;Unable to read and speak English and Dutch.

Three women were included in this pilot study, and their characteristics are shown in Table 1. For each of them, a different number of acquisitions (from 6 to 9) lasting 2–3 min was performed. All participants signed informed consent, and the experiment was approved by the ethical committee of the MMC in Veldhoven.

Furthermore, the dataset employed for the “Noninvasive Fetal ECG—The PhysioNet Computing in Cardiology Challenge 2013” (DS-PN) [19,20] was used. The dataset contains the signals employed to train and design the algorithm proposed in [16]. These multi-channel signals consist of 4 abdominal recordings acquired with contact electrodes and from multiple subjects using a variety of instrumentation with differing frequency response, resolution, and configuration. A total of 75 recordings were included in the dataset, each of them being 1 min long. For each acquisition, the reference position of the fQRS, and therefore the true fHR, is also available obtained from a fetal scalp electrode.

### 2.3. fHR Estimation

The method to estimate the fHR from dry electrodes was based on a hybrid pre-processing stage that combined hardware and software strategies to remove the artifacts introduced by the employed textile electrodes, followed by the main processing steps. In these steps, BSS methods were applied in cascade to reduce noise and artifacts and to identify the source corresponding to the electrical activity of the fetal heart. The main processing steps consisted of the detection of mQRS, the canceling of the mECG, and the enhancement of the fECG to identify the fQRS. The two main parts of the proposed approach were independent of each other; therefore, there was no link between the pre-processing and the main processing. Indeed, the pre-processing aim was to obtain artifact-free and noise-free signals to the utmost extent in order to sustain robust main processing and to obtain reliable fHR estimation.

Our novel algorithm was implemented in Matlab 2019b (MathWorks Inc., Natick, MA, USA) and modified the VA method that was developed and validated on standard wet recordings only. Substantial contributions were introduced in the pre-processing stage, and in the main processing, as highlighted in Figure 2. The stages that remained unchanged are reported in blue, while the introduced modifications are highlighted in red. From the scheme in Figure 2, it is possible to observe that each of the main steps was deeply modified to guarantee the robustness of the algorithm against the presence of artifacts.

In the main processing, the blind source separation (BSS) technique [21] employed to isolate the mECG and fECG made real-time processing unfeasible. For this reason, the acquired signals were stored in a buffer with a duration of one minute, without overlap. When the buffer was full, the entire processing could be performed. Samples from each unipolar channel were stored into the vectors $x_{i}={\left[x\left(T_{s}\right)\ldots x\left(NT_{s}\right)\right]}^{T}$, where i=1,…,4, *N* = 60,000 and $T_{s}=1/F_{s}$ was the sampling interval.

#### 2.3.1. Hybrid Pre-Processing

In this section, the strategies adopted to remove artifacts from the adopted textile electrodes are described in detail. This part, reported in Figure 2, was not present in the original approach VA. The proposed approach is based on the synergy between hardware and software, involving a hybrid approach to reject noise and artifacts.

##### Triboelectricity Artifacts

Lateral movement produces friction at the electrode–body interface and thus induces triboelectricity, which can consist of the sudden transfer of charge from the skin to the sensor. This type of artifact, amply described for capacitive sensors [22], can also be observed by the adopted textile electrodes, as the high resistance of the conductive fabric can still allow for charge accumulation. The occurrence of triboelectricity artifacts is unpredictable, and its amplitude is greater than a few units of Volt [22]; therefore, it can induce saturation of the channel. The recovery from saturation could easily last for tens of seconds, due to the required low high-pass corner and thus a large time constant. During this period, all the information would be lost, as shown in Figure 3a. To avoid this, the hardware was equipped with a reset circuit as in [23,24]. Figure 4 shows the block diagram of the hardware system built for this project. It can accommodate 4 channels, and each channel comprises several amplifiers, a low-pass filter and an analog-to-digital converter (ADC). A Field-Programmable Gate Array (FPGA) board transmitted the output data to a computer for further processing. The reset circuit was located in the first amplifier, which was activated when the amplifier (and thus also the ADC) is near the limit of the input range. This “reset” demand was generated as a reset pulse by the FPGA when it detected a channel’s ADC output close to the saturation thresholds (one near the high rail and another near the low rail). The four channels were controlled individually. When a reset occurred, the corresponding channel was forced to “reset” to the central value of the full-scale. In this way, the system could get rid of saturation and the long periods of information loss. On the other hand, this caused a step $x\left(T_{n}\right)-x\left(T_{n-1}\right)$, where $x(T_{n})$ is the sample value after reset and $x(T_{n-1})$ is the sample value before reset, as shown in Figure 3b.

The generated reset pulse indicated when and for which channel the reset occurred to perform the digital reconstruction: the difference between the average value of the signal post-reset and pre-reset was subtracted from the signal segment following the reset time. In this way, the DC component became constant over the full trace. Reconstruction is fundamental because it helps to recover the raw signal back from the broken pieces and thus restores the information. Although the triboelectricity artifact was not compensated for, the un-reconstructed signal would present a huge artifact that could prevent the correct estimation of the fHR. The lower plot of Figure 3c shows the un-reconstructed signal after the filtering, which will be described in the following paragraph.

##### Filtering

A notch forward-backward filter was applied to remove the power-line interference. It had zero phases and a bandwidth of 1 Hz. This was followed by a bandpass filter to remove the disturbances outside the frequency range of fetal QRS. The cut-off frequencies and the orders were selected to keep the shape and the amplitudes of the fECG unchanged. The adopted low-pass filter was a finite impulse response (FIR) filter with a cut-off frequency equal to 75 Hz and order 200. The high-pass filter was a Butterworth infinite impulse response (IIR) filter with order 6 and a cut-off frequency equal to 10 Hz.

##### Amplitude Demodulation

The presence of artifacts is dominant in real-life acquisitions with textile sensing due to variations in the sensor geometry and impedance induced by motion [25]. This introduces motion artifacts within the frequency band [0.5–120] Hz as well as an amplitude modulation of the signal affecting all the components of the acquired signal [26]. Similar artifacts can also be observed in capacitive sensing due to variation in the capacitor geometry [7]. Since these artifacts are superimposed both in time and in frequency on the fECG, they cannot be removed by standard filtering. The amplitude modulation could affect only a subset of channels; therefore, the demodulation was performed independently for each channel with the following steps:On the filtered signal $x_{i}$, where i=1,…,4, we detected the positive and the negative peaks of the QRS complexes by means of a peak detector based on adaptive thresholding. The detector in [27] adopted a finite state machine (FSM) to adapt the threshold to the amplitude variation of the signal induced by the modulation.Positive ($\gamma_{i+}$) and negative ($\gamma_{i-}$) envelops were calculated by interpolation of the peaks using a cubic spline.The envelops were combined as: $\Gamma_{i}=\left(\gamma_{i+}-\gamma_{i-}\right)/2$.A scale factor, defined as $\sigma_{i}=\mathrm{median}\left(\Gamma_{i}\right)$, was computed over the full trace to preserve the amplitude information.The de-modulated signal ${\hat{x}}_{i}$ was obtained according to: ${\hat{x}}_{i}=\left(x_{i}\cdot \sigma_{i}\right)/\Gamma_{i}$.

Figure 5 reports the comparison between the signals before (Figure 5a) and after (Figure 5b) compensation of the amplitude modulation.

#### 2.3.2. Main Processing

Motion artifacts cannot be effectively removed from the acquired signals; therefore, the main processing steps were designed to reduce their influence and to perform a reliable and accurate estimation of the fHR even if the number and the amplitude of the artifacts were high. In the main processing, BSS methods were applied in cascade to identify the source corresponding to the electrical activity of the fetal heart. In detail, this step involved the detection of the mQRS, the separation and canceling of the mECG, the enhancement of the fECG and the identification of the fQRS, as already performed in [16]. Our contribution consisted of the adaptation and improvement of this method for processing signals acquired by dry sensors. First, bipolar measurements were obtained by subtracting the pre-processed unipolar signals. Signals were considered in pairs, according to the arrows of Figure 1c. Four couples were considered:(1)$$y_{1}={\hat{x}}_{1}-{\hat{x}}_{4};\;y_{2}={\hat{x}}_{2}-{\hat{x}}_{4};\;y_{3}={\hat{x}}_{3}-{\hat{x}}_{4};\;y_{4}={\hat{x}}_{1}-{\hat{x}}_{3}$$

Each bipolar measurement $y_{i}$ with i=1,…,4 was a row of the matrix Y(4×N), where *N* is the number of samples acquired during one-minute acquisition.

Figure 6 reports the block diagram of the main processing stage. The diagram is divided into three main parts according to the main steps of the processing: the mQRS detection, the mECG canceling and the fHR estimation.

##### mQRS Detection

mECG is the predominant component of the signals and masks the fECG. Therefore, efficient canceling of the mECG is necessary. In order to extract the mECG, the separation of the signals into their main components was done by means of Independent Component Analysis (ICA), by applying the algorithm fastICA [21]. This algorithm is deflationary, i.e., the components are estimated in sequence, and the adopted cost function was the kurtosis [28]. In this way, the 4 main components of the signals were obtained: fastICA(Y)→Z, where Z has the same size as Y and each row $z_{i}$ with i=1,…,4 is one of the independent components.

According to the VA method, mECG canceling required that each cardiac cycle was approximated by singular value decomposition (SVD). Therefore, the signal was segmented into cycles by knowing the position of the mQRS. Detection of mQRS in VA was performed by selecting a priori the best component $z_{i}$ containing the mECG, i.e., the component that maximized a quality index based on the pseudo-periodicity of ECG signals (see [16] for further details). After the selection, the maternal QRS detection was performed only on the selected ICA component. This approach might sometimes identify the wrong component resulting in incorrect detection of the complexes and, therefore, ineffective canceling of the mECG. For example, in Figure 7, the component selected a priori by the algorithm was the first one. Therefore, aiming at more robust denoising, we proposed an a posteriori approach. This is detailed as follows:All the principal components were forward-backward filtered with a Butterworth bandpass filter (6.3–16 Hz) to enhance the QRS complexes, whose position was then identified by means of a peak detector based on adaptive thresholding [29], already introduced in [16]. Figure 7 shows the 4 main components, obtained with ICA, with the related QRS complexes identified by the peak detector. After this step, for each component the estimation of the maternal heart rate (${\mathrm{mHR}}_{i}$, with i=1,…,4) was derived as:
(2)$${\mathrm{mHR}}_{i}=60\cdot \left[1/\left(t\left({\mathrm{mQRS}}_{i,2}\right)-t\left({\mathrm{mQRS}}_{i,1}\right)\right),\ldots ,1/\left(t\left({\mathrm{mQRS}}_{i,nq-1}\right)-t\left({\mathrm{mQRS}}_{i,nq}\right)\right)\right],$$
where ${\mathrm{mQRS}}_{i,j}$ with $j=1,\ldots ,N_{QRS}$ is the sample of the *j*-th R peak and nq is the total number of identified QRS complexes.The series, and related component, that best represents the maternal HR was selected a posteriori according to the following procedure: for each component, the mean value of the maternal heart rate series was computed; if this value was outside the range [50–180] beats per minute (bpm), the series (and related component) was discarded. If all series were discarded, the mECG removal was not performed. For the remaining series, the following parameters were derived to select the best candidate:Number of outliers $N_{outliers}$, i.e., number of the series elements outside the range 50–180 bpm;Series variability, defined as the sum of the absolute difference between successive elements of the series. This criterion is based on the pseudo-periodicity of the ECG signal [30], since we expected the HR to show limited variability between subsequent beats.The discrepancy of the mean value of the series with a predefined mean value of the HR, $\bar{mHR}=85$ bpm. This criterion was adopted to avoid confusion between fECG and mECG.These parameters were combined by the following formula to establish a quality index for the selection of the best candidate component:
(3)$$Q=\frac{N_{outliers}}{N_{QRS}}+\frac{\sum_{j=1}^{N_{QRS}-1}\left|mHR_{j+1}-mHR_{j}\right|}{\sum_{j=1}^{N_{QRS}}mHR_{j}}+\frac{\sum_{j=1}^{N_{QRS}}\left(mHR_{j}-\bar{mHR}\right)/N_{QRS}}{\bar{mHR}}$$In Figure 7, the selected mHR series was the one related to the second independent component, which seems correct.

**Figure 7 sensors-21-04298-f007:**
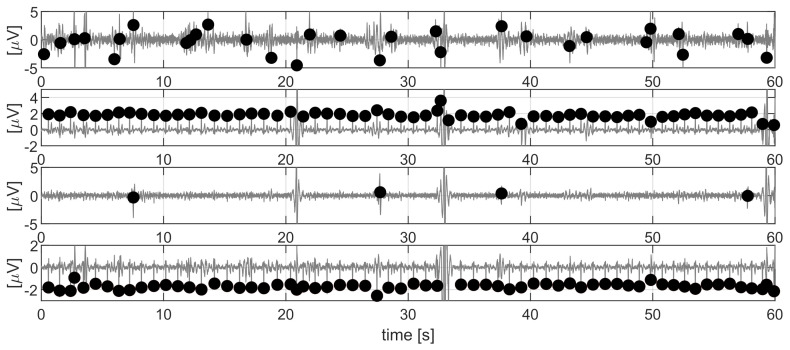
The main independent components obtained after the first ICA and the mQRS (black dots) detected for each component.

##### mECG Canceling

mQRS complexes were used for signal segmentation. Even if the selected series was related to one component only, the position of the related mQRS was employed to segment all the independent components. Each component (row of the Z matrix) was divided into cardiac cycles, which were centered around a mQRS and weighed by a trapezoidal window that avoided abrupt artifacts due to truncation. Each cycle became a column of a matrix K of size $N_{d}\times N_{q}$, where $N_{d}$ is the length of the segment and $N_{q}$ is the number of mQRS. K was decomposed by SVD:(4)$$K=UHV^{T},$$
where H ($N_{q}\times N_{q}$) is a diagonal matrix of singular values, U ($N_{d}\times N_{q}$) and V ($N_{q}\times N_{q}$) are the unitary matrices of the left and right singular vectors, respectively. Because the columns of the matrix K consist of synchronized and weighted cardiac cycles, the maternal cardiac waves, being the signal element with larger amplitude, should produce the strongest contribution to covariance. Therefore, the first left singular vectors, i.e., first columns of the matrix U, which correspond to the first principal components, mainly represent the maternal cardiac cycle. The remaining components should be related to the other signals, such as fECG, noise and artifacts. Therefore, only the first major components should be used to obtain an approximate version of the mECG.

However, if there are large amplitude artifacts, as shown in Figure 8, where the signals were acquired with our-textile electrodes, the first SVD components contained information about the artifact and the information about the mECG was split into the other components. Due to this dispersion, the approximated mECG underestimated the real one; Figure 8a shows the maternal cardiac cycle template obtained by averaging all the approximated segments, also those containing artifacts, such as the purple one in the figure. When the approximated version of the mECG was subtracted, the mECG was canceled partly or not at all, as shown in Figure 8c.

To overcome this problem, the proposed approach introduced some improvements to VA [16]. The improvements consisted of identifying the segments containing artifacts and discarding them from the reconstruction. We proposed to define for each cardiac cycle the maximum value of an artifact-free R peak as 5 times the median value of all the peaks. Then, all the segments with a maximum value higher than this threshold were discarded.

Matrix $K_{s}$, and the related SVD decomposition ($K_{s}=U_{s}H_{s}V_{s}^{T}$), were based only on the segments with the cardiac cycles not affected by artifacts. In Figure 8b, we can observe that the resulting template soundly approximates the shape of the mECG. $K_{s}$ has size $N_{d}\times \left(N_{q}-N_{s}\right)$ where $N_{s}$ is the number of discarded columns and the other matrices have size according to $K_{s}$. Matrix $K_{s,r}$ was reconstructed in order to contain only maternal cardiac cycles.
(5)$$K_{s,r}=U_{s,r}H_{s,r}V_{s,r}^{T}$$
where $H_{s,r}$ ($N_{e}\times N_{e}$) is a diagonal matrix with only the first $N_{e}$ singular values, $U_{s,r}$ ($N_{d}\times N_{e}$) and $V_{s,r}$ ($N_{q}-N_{s}\times N_{e}$) are the matrices with the first $N_{e}$ left and right singular vectors, respectively.

The approximated cardiac cycles, i.e., the columns of the matrix $K_{s,r}$ were connected with a straight line, and the signal portions corresponding to the $N_{s}$ segments previously discarded were replaced with a straight line. In this way, the estimated mECG was obtained ($m_{i}$, i=1,…,4) and it correctly approximates the real one, which was effectively removed by subtraction, as shown in Figure 8d. Given the matrix M that contained in each row the approximated mECG of the related component ($m_{i}$, i=1,…,4), the signals without the mECGs were given by: W=Z−M.

##### fECG Enhancement and fHR Estimation

Based on VA [16], the same fastICA algorithm, just employed to enhance the mECG, was applied on the matrix W to obtain the independent components of the residual signals. Detection of the fetal QRS complexes was performed independently on all four signals and was based on a derivative filter that enhanced the QRS complexes. Then, on the filtered signals, a detector was applied (det. 1 in the scheme presented in Figure 2), which was based on the same principle as the algorithm employed to identify the mQRS. For each component, the position of the fQRS complexes was used to estimate fHR “good” segments, which were defined as the longest interval whose fHR values were close to the mode of the fHR series of the corresponding component. The VA method proceeded by analyzing the independent components individually and independently: for each of them, the fHR values estimated in the “good” interval were used to initialize a second detector (det. 2 in the scheme reported in Figure 2) of the QRS complexes, which was applied in the forward/backward direction, from the beginning and the end of the interval, respectively. The second detector employed an autoregressive (AR) model, implemented through an adaptive filter, to predict the position of the next fQRS. The detector looked for the maximum of the derived signal, which was weighted by a trapezoidal window that enhanced the samples close to the predicted position of the next QRS.

The filter coefficients were estimated on the initial interval and were updated every time a new QRS complex was detected. fQRS detection was applied to each component independently of the others, obtaining four hypothetical QRS annotations and the corresponding fHR series. Finally, the best fHR series was selected.

This approach, however, was not robust enough to artifacts. It was based on the assumption that, after the filtering and canceling of the mECG from the signals, the number of remaining sources is less than or equal to four, and therefore, the ICA enables to separate and enhance the fECG. This assumption was not respected for signals acquired with dry electrodes. Indeed, artifacts that corrupt the signals acquired with dry electrodes were more and have greater amplitude than the artifacts of signals acquired with adhesive electrodes. Each artifact was considered by ICA to be an independent source; as the total number of sources was ≥4, the fECG cannot be isolated.

For this reason, the final part of the proposed algorithm deviated from VA, as depicted in Figure 9. Among the 4 “good” intervals, one for each component, the optimal one ($S_{opt}$) was selected according to the following procedure: if the correlation between the position of the fQRS complexes and the mQRS was greater than 70% or the mean value of the fHR was outside the range 100–180 bpm, the interval was discarded, as it is recognized as an artifact or maternal residual. Among the remaining intervals, the longest was chosen, as we considered length as a synonym for the quality of the estimation. The fHR values estimated over this interval were employed to initialize a second detector (det. 2), which was the same described in the previous paragraph.

The column elements of W corresponding to the selected interval are called $S_{opt}$ in Figure 9. Parts of W before and after the $S_{opt}$ were divided into segments of 10 s with 5 s of overlap with the consecutive segment, and ICA was applied to each segment. The length was set to be large enough to get a reliable estimation of the components by ICA and short enough to reduce the number of motion artifacts for each segment. Since reducing the length, the number of artifacts in each segment was limited, the number of sources was lower than four, and the fECG could be separated. The segments before and after $S_{opt}$ were namely $S_{b}$ and $S_{f}$, respectively. The first segments in both the directions ($S_{f,1}$ and $S_{b,1}$) were partially overlapping $S_{opt}$, as shown in Figure 9.

fHR estimation was performed by means of the second detector, which was applied both in forward and in backward directions starting from the optimal segment. The detection of the fQRS proceeded in blocks according to the following procedure described for the generic block *k*: the *k*-th block $b_{k}$, lasting 5 s, was part of both the segments $S_{k}$ and $S_{k+1}$. For each block, there were eight independent components obtained by two ICAs, which were each applied to a different segment producing four components. The number of components was doubled by employing the overlap to increase the robustness even further (more components lead to a greater probability of separating the fECG). For each of the eight components, the fHR was estimated ${\mathrm{fHR}}_{k,h}$, h=1,…,8. The eight series were added at the end of the fHR series previously estimated (${\mathrm{fHR}}_{h-all}=\left[{\mathrm{fHR}}_{opt},{\mathrm{fHR}}_{1},\cdots ,{\mathrm{fHR}}_{k-1},{\mathrm{fHR}}_{k,h}\right]$, h=1,…,8). The selection of the best fHR estimation for the considered block (${\mathrm{fHR}}_{k}$) was made considering the whole series estimated up to that point (${\mathrm{fHR}}_{h-all}$), according to an index based on the following characteristics:Series variability based on the mean of the absolute first and second derivative of the HR series values. This constraint was based on a priori knowledge of the fHR regularity.The correlation between the position of the fQRS and the mQRS, introduced to avoid the selection of the maternal series in the presence of the mECG residual left.If the mean value of the fHR was outside the range [100, 180] bpm, a penalty factor was added and depended on the deviation of the mean value from the edges of the range.

The algorithm performed the analysis block by block until the end of the trace. The final fHR estimation was obtained by combining the estimation of the best fHR related to each block, which corresponded to the best main component of the block.

In this novel approach, all the components were used for the estimation of the fHR. Indeed, all the elements were used and recombined at each step, contrary to VA, where only one independent component was selected. In the proposed method, all the information was preserved and the motion artifacts were distributed over several segments in order to reduce their influence on the separation of the fECG. This method enables creating a more robust system that provides an accurate and reliable fHR estimation even for a large number of artifacts.

### 2.4. Evaluation Metrics

The performance of the fHR detection algorithms was quantified using the reference annotations available. Each beat was classified as True Positive (TP) or False Negative (FN) if the actual fQRS complex was detected or missed, respectively. Instead, False Positive (FP) were points that were identified as fQRS but actually were not. A QRS complex was considered correctly identified if the estimated position differs less than 50 ms from the reference annotation, according to the guidelines reported in [31].

Sensitivity and F1-score indices were employed to evaluate the ability of the algorithm to correctly detect fQRS complexes:Sensitivity (Se) measures the proportion of actual fQRS complexes that are correctly identified as such:
(6)$$\mathrm{Se}=\frac{\mathrm{TP}}{\mathrm{TP}+\mathrm{FN}}\cdot 100$$F1-score measures the overall performance of the algorithm to identify fQRS complexes:
(7)$$F1-\mathrm{score}=\frac{2\cdot \mathrm{TP}}{2\cdot \mathrm{TP}+\mathrm{FN}+\mathrm{FP}}\cdot 100$$

Then, the accuracy and the reliability of the provided fHR estimation were evaluated according to the following metric:Root Mean Square Error (RMSE) measures in beats per minute (bpm) the difference between the true fetal HR (fHR) and the estimation provided by the algorithm ($\hat{\mathrm{fHR}}$):
(8)$$\mathrm{RMSE}=\sqrt{\frac{\sum_{l=1}^{P}{\left({\hat{\mathrm{fHR}}}_{l}-{\mathrm{fHR}}_{l}\right)}^{2}}{P}}$$
where *P* is the number of heartbeats of the record.

### 2.5. Statistical Tests

To compare the results obtained with the two approaches, a statistical test was performed for each metric in order to recognize significant differences. The normality of the sample distributions was tested by the Kolmogorov–Smirnov test [32]. Then, we selected the *t*-test or the Wilcoxon rank statistical test, according to the normality or the non-normality distribution of the samples, respectively. Test results were considered significant for *p*-values lower than 0.05.

## 3. Results

For the data sets DS-PN and DS-SS, the reference annotations are available; therefore, a quantitative performance analysis is feasible. Table 2 and Table 3 report the mean value, with the related standard deviation, the median value, and the interquartile range of the metrics described in Section 2.4 evaluated, respectively, on DS-PN and DS-SS with VA and the proposed methods. Figure 10 and Figure 11 are related to DS-PN and DS-SS, respectively, and report a few examples of the fHR estimated both with VA and the proposed method. Figure 12 shows the boxplot of the F1-score (Figure 12a) and RMSE (Figure 12b) metrics obtained on the DS-PN and DS-SS with both VA and the proposed methods.

Because the acquisitions of DS-R are real, a reference fECG signal is not available; therefore, the performance on this database cannot be evaluated in a quantitative way, but only a visual qualitative analysis is possible. Figure 13 shows one example of the estimation of the mHR and fHR, by means of VA and the proposed method, for each participant in this pilot study.

## 4. Discussion

In this work, we propose a method to detect fHR from multi-channel abdominal recording acquired with dry textile electrodes. The lack of gel between the skin and the electrode increases the contact impedance, which results in lower signal amplitude even when employing high input-impedance amplifiers, making the extraction of the fHR extremely complex. Furthermore, due to the lack of contact, the electrodes employed in this work are subject to vertical and lateral movements with respect to the body surface, which significantly reduce the quality of the acquired signals. Therefore, the method is specifically designed to remove disturbances typical of these recordings, and it is based on a hybrid (hardware–software) pre-processing stage and main processing steps. The pre-processing removes triboelectricity artifacts and amplitude modulations. On the other hand, the main processing stage consists of the removal of the maternal ECG by blind source separation, the enhancement of the fetal ECG and identification of the fetal QRS complexes. The main processing is modified with respect to VA to be more robust to the high amplitude motion artifacts that corrupt the acquisitions with dry electrodes.

The proposed method produces results comparable or even better than the approaches presented in the literature for the estimation of fHR from signals acquired with wet electrodes, even if our algorithm was not specifically designed for this type of acquisition and it is more suitable for acquisitions performed with dry electrodes. For instance, the denoising algorithm based on PCA presented in [12] reached a sensitivity of 95% and an RMSE lower than 5 bpm, which are comparable with the performance of the proposed method in the traces of DS-PN, shown in Table 2. Instead, the proposed method outperforms the results obtained by [15] as this method, based on the combination of ICA and EMD, produced F1-score and sensitivity lower than 90%. However, these results can only provide a rough indication of the performance, as they are obtained on different databases, acquired with different instruments and in different stages of pregnancy. Therefore, we compared our method with VA on equal datasets, acquired both with wet and dry textile electrodes.

For what concerns the results obtained considering the traces of DS-PN, the proposed algorithm has a performance comparable with VA. Despite the modifications introduced, the proposed algorithm is still suitable for multi-channel acquisitions performed with contact electrodes. From Table 2, we can see that the performance obtained with the two methods is not significantly different for any of the considered metrics.

In the last stage of the proposed method, segmenting the signals could induce estimation errors at edges due to the truncation of the QRS complex, which involves a missing beat. An example of this event is related to the a19 record, shown in Figure 10a, where estimation errors are present at 28, 42 and 49 s. Nevertheless, the proposed method is useful for signals acquired with contact electrodes that are exceptionally affected by high amplitude motion artifacts. For example, in record a34, shown in Figure 10b, there is an artifact in the final part of the acquisition that involves an error in the fHR estimation by VA (upper), which is not present when the proposed method is employed (lower).

The proposed method outperforms the results obtained by VA on the DS-SS (Table 3) and the differences, according to the obtained *p*-value, are significant for all the considered metrics. In detail, sensitivity has improved from 77.0% to 89.7%, the mean F1-score value from 81.9% to 92.8% and the mean RMSE is almost 13 bpm lower. VA provides a worse outcome for all the traces considered, due to its lack of robustness to artifacts that are specific to recordings making use of dry electrodes. An example of a trace in which the estimation provided by both the algorithms is representative of the average behavior is shown in Figure 11a. Figure 11b is related to one of the recordings with fewer artifacts, VA (upper plot) does not provide an accurate estimation when artifacts are present, and on the other hand, the estimation provided by the proposed method is correct even in the presence of artifacts according to the RMSE error lower than 1 bpm. Figure 11c shows a case where the performance of the two approaches is comparable (RMSE =1.7 bpm for the proposed approach and 3.1 for VA).

It can be noted that the performance of the algorithms is polarized, i.e., metrics values are very low or very high, without intermediate values, as shown in Figure 12, where the boxplots of F1-score and RMSE values are reported. Therefore, when the algorithm is unable to accurately denoise the multi-channel acquisition, the provided estimation of the fHR is completely erroneous and the related F1-score and RMSE values are very low and very high, respectively. For example, in Figure 11d, one of these cases is shown: VA (upper plot) provides a completely erroneous estimation of the fHR, to which the F1-score =50% and RMSE =37.8 bpm are associated. In contrast, the proposed method provides a more accurate result (F1-score =85% and RMSE =11.6) but is still not reliable due to the very low SNR (<−60 dB), compared to the other recordings. Since the values are not normally distributed, it is useful to consider also the median and the interquartile range to describe and analyze the results.

The median values are generally higher than the mean values, in particular for VA and the data set DS-SS. Therefore, the difference between the performances obtained with VA and the proposed method is reduced. For what concerns the RMSE metric, the mean value obtained on DS-SS is 19.4 bpm and reduces to the median value of 8.6 bpm for VA. However, for the proposed method, the reduction is from 6.6 to 3.1. Instead, the median value of the sensitivity for VA is only 88.6%, against the 100% of the proposed method. This means that VA misses a high number of fQRS complexes. Furthermore, the interquartile ranges of all the considered metrics are at least three times larger for VA than for the proposed method. This means a high variance and dispersion of the estimations provided by the method [16], which does not show consistent performance over all the traces. This behavior confirms the greater robustness of the proposed method compared to VA. Indeed, the proposed method is able to provide an accurate estimation of the fHR even in the presence of very noisy signals and with very low SNR: the value of the RMSE is less than 8 bpm in 70% of the subjects, which reduces to only 40% of the subjects for VA.

Finally, the algorithms are employed on the real multi-channel recordings of DS-R. In this case, the reference annotations are not available; therefore, only qualitative analysis based on physiological knowledge from the literature can be provided [33]. As can be seen from Figure 13, the estimations of the mHR obtained by VA and the proposed method coincide for all the acquisitions, and the values are in a range considered physiologically acceptable. In addition, the fHR estimates in Figure 13b almost coincide, even if that obtained by VA is slightly more variable.

Instead, in Figure 13a,c, only the estimations obtained with the proposed method are within the physiological range of the fHR, i.e., 120–160 bpm, when the women are not in labor. The estimation by VA in Figure 13c is approximately 180 bpm, which is too high for a 39-week fetus, while in Figure 13a the estimated fHR is lower than 100 bpm and, moreover, the series is characterized by high variability, which is not compatible with the heartbeat and suggests an estimation error.

In this study, dry, textile electrodes were employed for the measurements. These high-resistance electrodes produce triboelectricity artifacts due to the charge and discharge of large potentials that are comparable to capacitive recordings. Motion artifacts due to changes in the sensor geometry may also recall typical artifacts of capacitive sensors due to vertical motion inducing variations in the sensor capacitance. Therefore, although specific validation making use of capacitive sensors is required, the proposed hybrid solution to deal with triboelectricity and the following pre-processing strategy may also represent a valid approach for capacitive recordings.

The main limitation of the proposed work consists of the small number (3) of participants in our pilot study. Furthermore, the lack of a reference for the DB-R database limits the evaluation of the algorithm performance to qualitative analysis, precluding objective quantitative analysis. In future studies, the availability of a reference signal, based, for example, on a scalp electrode, could be considered. Finally, the data used in this work were acquired by placing the textile electrode on the skin. In future developments, the proposed algorithm will also be tested and evaluated with textile layers.

## 5. Conclusions

To conclude, the results of the presented work confirm the need for developing a specific system to take on the challenges of unobtrusive fetal ECG recordings by dry electrodes, as the signals acquired with these electrodes are very different from those recorded with standard wet electrodes. Since the sources of noise that corrupt the signals are not the same and have different characteristics, these needed to be taken into account in the design of the algorithm in order to pursue an effective and suitable denoising. The proposed method is the first work in which BSS methods are employed combined with denoising steps specifically designed for signals acquired with capacitive instrumentation in order to suppress the artifacts introduced by the lack of contact between the electrode and the skin. When processing dry-electrode recordings, the proposed method outperforms the original methods designed for wet-electrode recordings, providing an accurate and reliable fHR estimation.

Future work will be needed to increase the number of electrodes employed so as to make the system more robust to the presence of motion artifacts. We will also investigate the influence of the electrode position on the performance of the proposed algorithm to identify the best electrode configuration. Finally, we plan to include more pregnant women in our study to obtain a more variable and heterogeneous dataset, which will enable us to test the generalizability of this approach. In addition, the availability of a reference for the fHR estimation assessment is a requirement, which will be addressed.

## Figures and Tables

**Figure 1 sensors-21-04298-f001:**
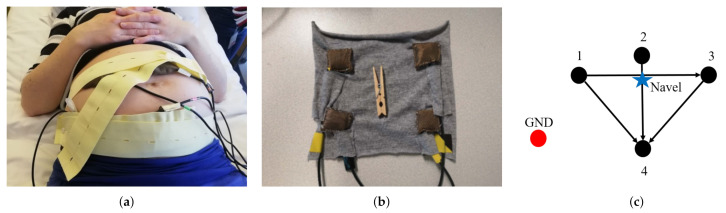
Features of the employed acquisition system. (**a**) Placement of electrodes by means of elastic belts on the mother’s belly. (**b**) Four textile electrodes. (**c**) Schematic configuration of the electrodes and the related bipolar leads, which are identified by arrows. The configuration is symmetric with respect to the navel.

**Figure 2 sensors-21-04298-f002:**
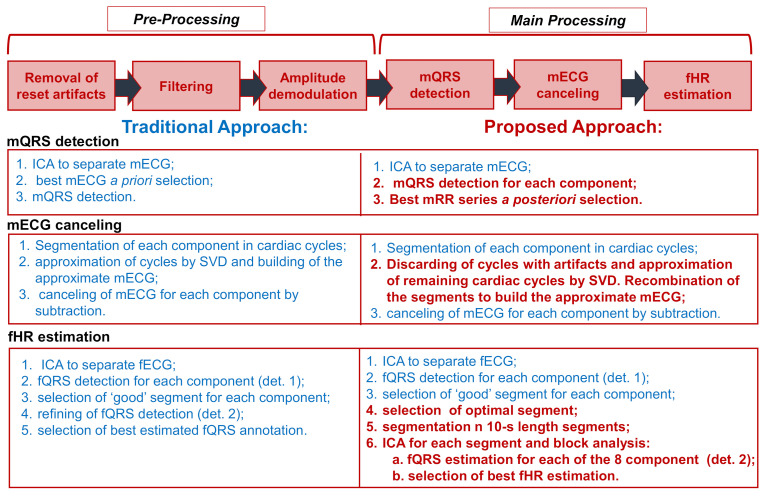
Comparison between the proposed approach and VA. Hybrid (hardware and software) pre-processing consists of the removal of triboelectricity artifacts, filtering, and amplitude demodulation and was introduced in the proposed approach to managing the artifacts typical of signals acquired with dry electrodes. Main-processing consists of three steps, which are the same as in VA, improved to guarantee robustness against artifacts. In the three lower blocks, the details of each step are reported both for VA and the proposed approach. In red, the modifications introduced are reported, and in blue, the stages that remain unchanged are reported.

**Figure 3 sensors-21-04298-f003:**
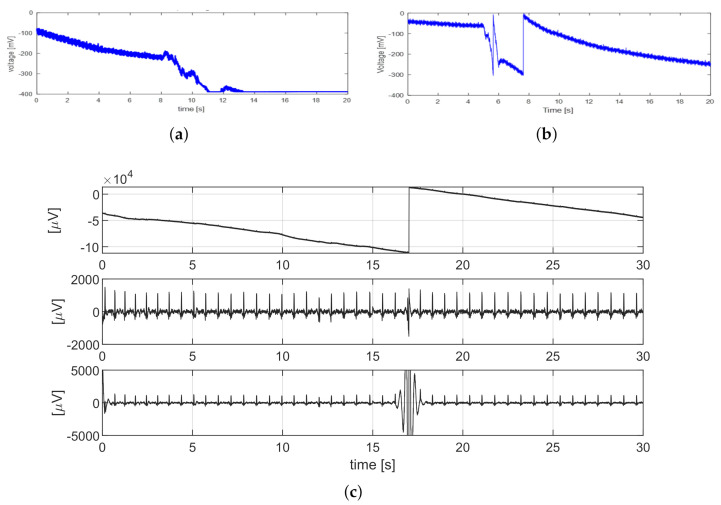
(**a**) Example of a channel acquired without reset. (**b**) Example of a channel acquired with reset. (**c**) Influence of the reset artifact on the signal. The plot above shows the acquired signal; at t=17 there is a step due to the reset. The signal after filtering if the step is removed is shown in the middle. The plot on the bottom shows the filtered signal if the step is not removed.

**Figure 4 sensors-21-04298-f004:**
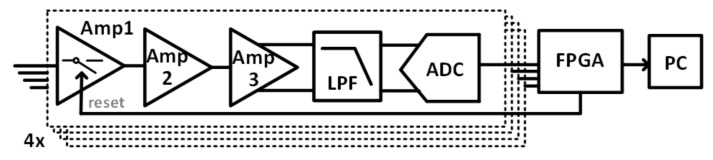
Block diagram of the hardware system employed in this project.

**Figure 5 sensors-21-04298-f005:**
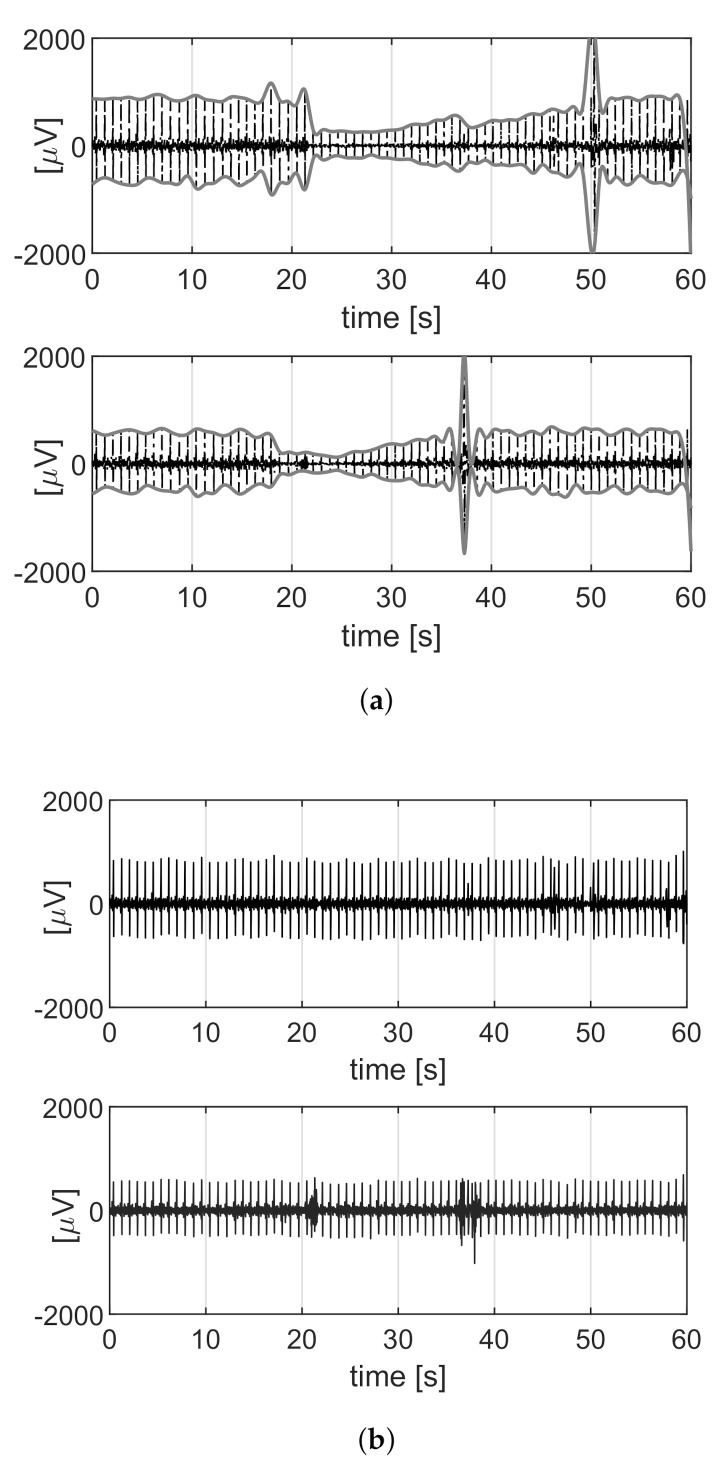
(**a**) Raw signals (black) affected by the amplitude modulation and the positive and the negative envelops (grey). (**b**) How the related signals appear after the compensation of the modulation.

**Figure 6 sensors-21-04298-f006:**
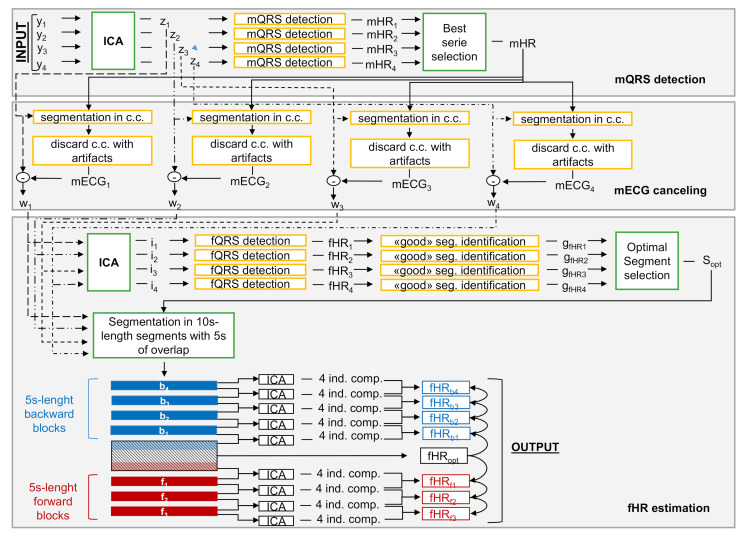
Block diagram of the steps (mQRS detection, mECG canceling and fHR estimation) of the main processing stage. The input is the matrix Y, which contains the bipolar measurements obtained after the pre-processing stage. The output is the fHR estimation. For the yellow blocks, only 1 element is involved; on the contrary, for the green blocks, all the elements are involved.

**Figure 8 sensors-21-04298-f008:**
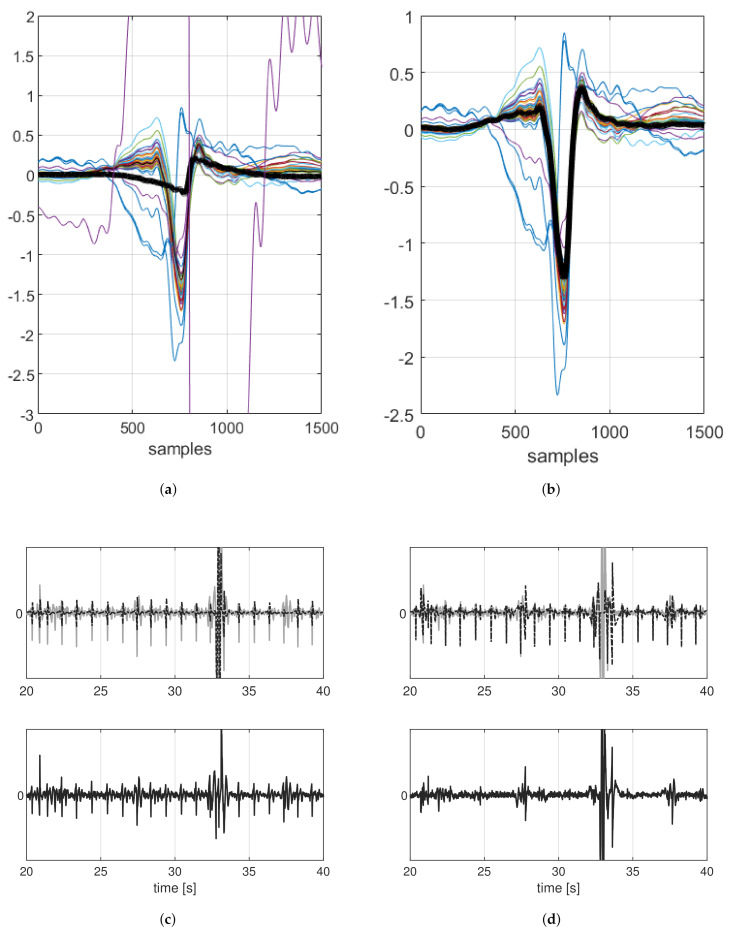
(**a**,**b**) Aligned segments employed for SVD decomposition and the related obtained template (black). (**a**) Approximation of the cardiac cycle is wrong due to the presence of high artifacts. (**b**) The estimation is correct by discarding the segments with artifacts. (**c**,**d**) Comparison between the independent components obtained by ICA (grey), the mECG estimated (black dash-dot) and the resulting signals after the canceling (black). (**c**) Canceling with VA, part of the mECG is not correctly removed. (**d**) Employing the proposed approach, the mECG is effectively removed.

**Figure 9 sensors-21-04298-f009:**
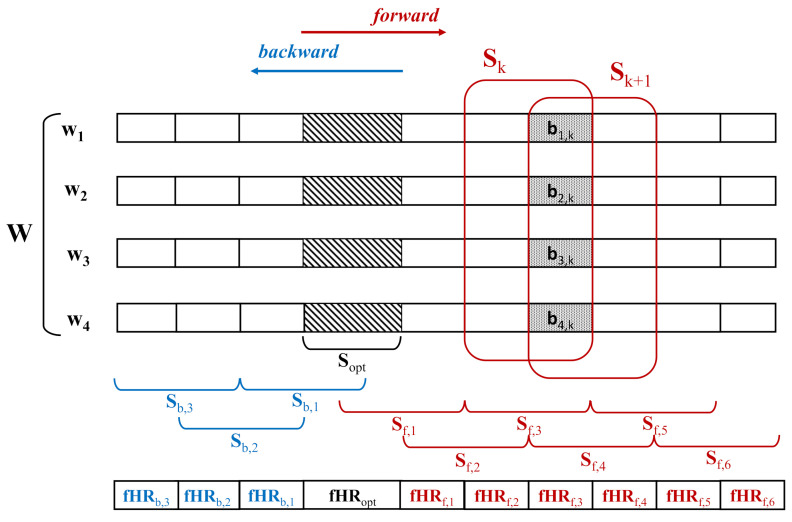
Signal segmentation scheme for the last step of the fHR estimation algorithm.

**Figure 10 sensors-21-04298-f010:**
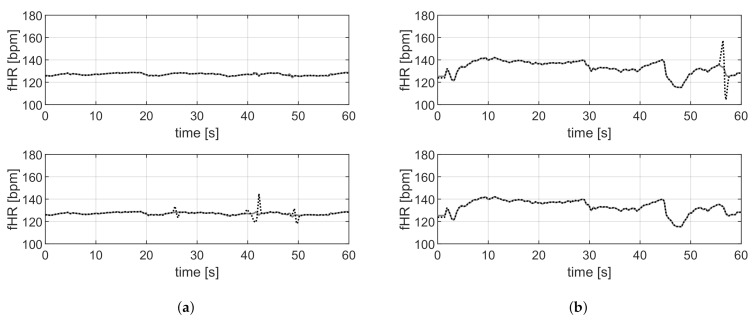
Examples of outcomes on DS-PN. Comparison of the fHR estimation with VA (upper) and the proposed method (lower), the reference is the gray line. (**a**) Record a19. (**b**) Record a34.

**Figure 11 sensors-21-04298-f011:**
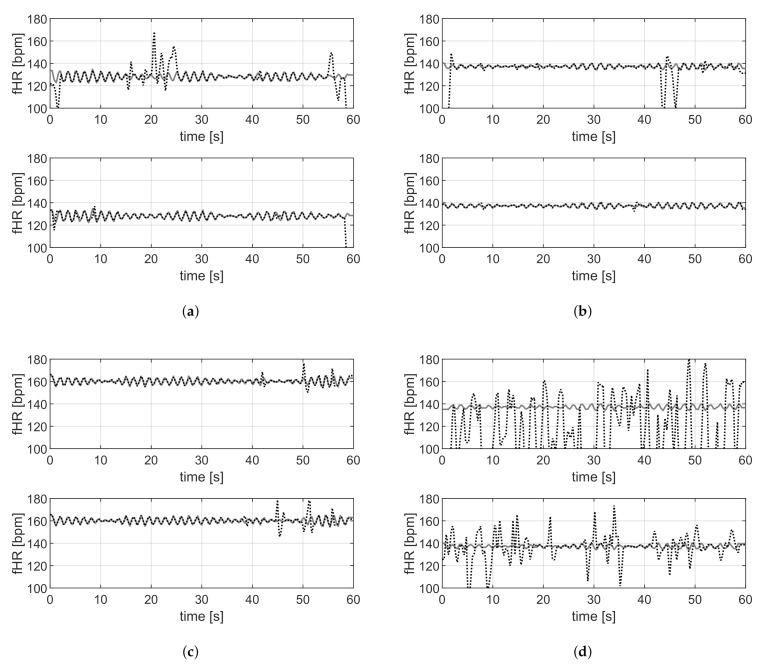
Examples of outcomes on DS-SS. Comparison of the fHR estimation with VA (**upper**) and the proposed method, (**lower**) the reference is the gray line.

**Figure 12 sensors-21-04298-f012:**
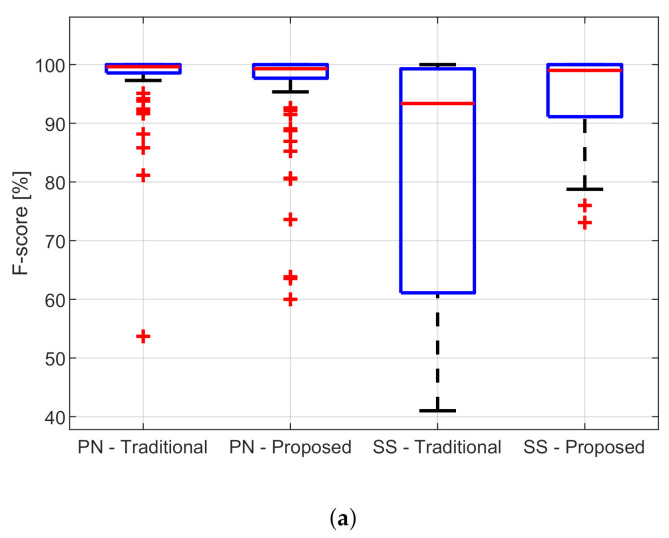
Boxplot of the results obtained on the DS-PN and DS-SS with VA and the proposed methods. (**a**) F1-score (**b**) RMSE.

**Figure 13 sensors-21-04298-f013:**
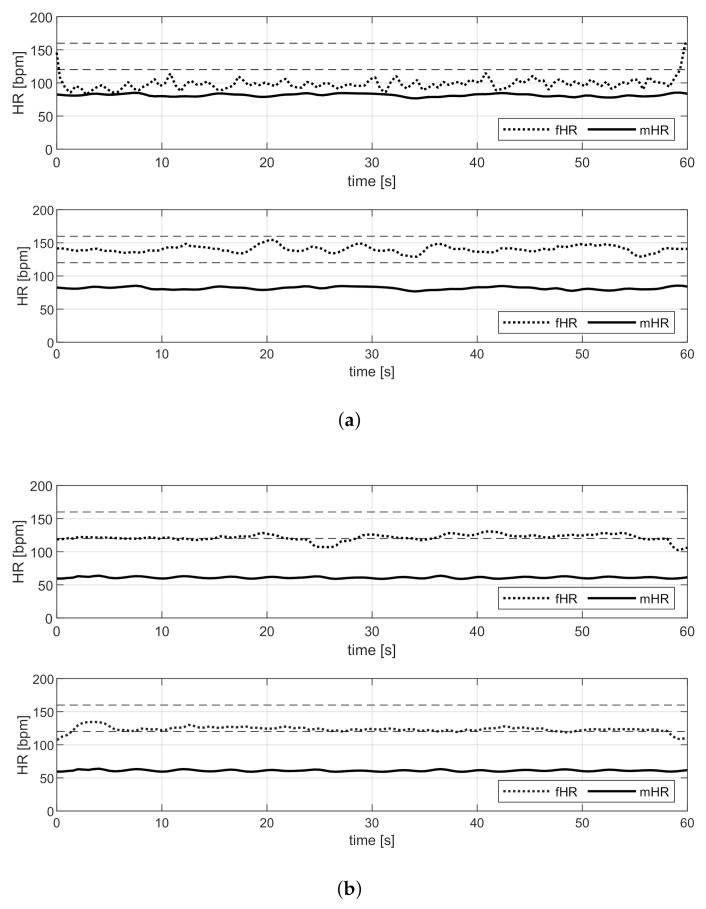
Examples of outcomes on DS-R. Comparison of fHR (dot) and mHR (line) estimation by VA and by the proposed method (upper and lower, respectively). (**a**) Real Acquisition 1. (**b**) Real Acquisition 2. (**c**) Real Acquisition 3.

**Table 1 sensors-21-04298-t001:** Characteristics of the women involved in the trial (DS-R). GA = gestational age (weeks, days); G = gravida; P = para; BMI = body mass index.

Sub-ID	Age	GA	G	P	BMI
001	32	38,0	4	3	27
002	31	36,4	2	1	21
003	36	39,2	2	1	23

**Table 2 sensors-21-04298-t002:** Mean ± standard deviation, median value, interquartile range (iqr) of the proposed metrics evaluated on DS-PN (Physionet Database) with VA and the proposed methods. To compare the methods, the *p*-value is reported. DS-PN dataset contains 75 subjects.

	VA [16]			Proposed Approach			
	Mean ± Std	Median	Iqr	Mean ± Std	Median	Iqr	*p*-Value
Sensitivity (%)	97.8±6.4	100.0	0.7	95.7±9.6	100.0	2.0	0.23
F1-score (%)	97.0±9.1	99.6	1.4	96.0±9.8	99.3	2.9	0.26
RMSE (bpm)	4.2±7.9	1.2	2.8	4.9±7.9	1.6	5.7	0.34

**Table 3 sensors-21-04298-t003:** Mean ± standard deviation, median value, interquartile range (iqr) of the proposed metrics evaluated on DS-SS (Semi-Simulated Database) with VA and the proposed methods. To compare the methods, the *p*-value is reported. DS-SS dataset contains 27 semi-simulated subjects.

	VA [16]			Proposed Approach			
	Mean ± Std	Median	Iqr	Mean ± Std	Median	Iqr	*p*-Value
Sensitivity (%)	77.0±26.0	88.6	46.9	89.7±15.2	100.0	15.8	0.03
F1-score (%)	81.9±21.2	93.4	38.1	92.8±11.6	99.0	10.3	0.02
RMSE (bpm)	19.4±23.5	8.6	30.8	6.6±8.6	3.1	8.2	0.04

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
