# Peer review of "Dedicated Algorithm for Unobtrusive Fetal Heart Rate Monitoring Using Multiple Dry Electrodes"

_sensors, 2021, doi:10.3390/s21134298_

Round 1

Reviewer 1 Report

Dear authors,

I would first congratulate you for your work - it is really important for the clinical practice. However, your manuscript needs some corrections:

Major:

  1. Please, make the introduction more succinct. Parts of its text sound like discussion and are more suitable for the “Discussion”.
  2. Please, add section “Statistical analysis” in “Materials and methods”. The methods you’ve decided to use should not be described (mentioned first) in “Results”.

  3. You have added "Inclusion criteria", which is very good. Please, add also some “Exclusion criteria” for your study.
  4. Please, remove all tables and the graphics, demonstrating examples of your results from the discussion. They should be moved to “Results”.

Minor:

1. Line 22: “pregnancy” should be “pregnancies”

2. Line 52: Please, add “a” before “subject”

3. Line 61: Please, define (describe) what “PQRST complex” is – it could be obvious to cardiologists, but other specialists may think it is an abbreviation, not spelled in advance.

4. Line 88 “…was a part of a larger project” sounds slightly better…

5. Lines 129-130 “…was based on” and  “…that combined…”seems more correct.  Please, keep to the past simple tense for the text, presenting your work (results) in the article

Author Response

Dear authors,

I would first congratulate you for your work - it is really important for the clinical practice. However, your manuscript needs some corrections.

We would like to thank the Reviewer for her/his positive feedback.

We have modified the manuscript according to the Reviewer’s observations as explained in more detail in the following. In the revised version, the modified parts are highlighted in red.

Major:

  •   Please, make the introduction more succinct. Parts of its text sound like discussion and are more suitable for the “Discussion”.

According to the reviewer’s comment, part of the introduction has been moved to the Discussion.

  •   Please, add section “Statistical analysis” in “Materials and methods”. The methods you’ve decided to use should not be described (mentioned first) in “Results”.

We added the "Statistical tests" subsection after the Section "Evaluation Metrics". In this Section, we defined the Statistical tests employed both to test the normality of the data and to quantify the significance of the differences between the VA and the proposed approaches. 

  •   You have added "Inclusion criteria", which is very good. Please, add also some “Exclusion criteria” for your study.

      We would like to thank the reviewer for raising this point. In the revised version of the manuscript we have included the exclusion criteria according to the following:

-  Age <18 years

- Pregnancy with maternal or fetal complications

- Unable to read and speak English and Dutch

  •   Please, remove all tables and the graphics, demonstrating examples of your results from the discussion. They should be moved to “Results”.

Only Fig. 13 was actually in Section “Discussion”, all the previous Figures and Tables were in Section “Results”. However, the layout could be confusing and make it appear that this was not. For this reason, we have moved Fig. 13 in Results and rearranged the layout to avoid this misunderstanding.

Minor:

  1. Line 22: “pregnancy” should be “pregnancies”
  2. Line 52: Please, add “a” before “subject”
  3. Line 61: Please, define (describe) what “PQRST complex” is – it could be obvious to cardiologists, but other specialists may think it is an abbreviation, not spelled in advance.
  4. Line 88 “…was a part of a larger project” sounds slightly better…
  5. Lines 129-130 “…was based on” and  “…that combined…”seems more correct.  Please, keep to the past simple tense for the text, presenting your work (results) in the article

Thank you for the careful reading of our manuscript. All the minor corrections were addressed in the new version.

Reviewer 2 Report

The manuscript describes a denoising system, consisting of a combined hardware and software solution, able to remove the disturbing component typical of acquisitions by dry electrodes and identify fQRS complexes. The performance of the proposed systems was evaluated in terms of sensitivity, F1-score, and root-mean-square error (RMSE) on signals selected from two datasets.

The manuscript is well written and organized. The results seem to be competitive in comparison with existing studies existing in the literature.

Some comments to improve the quality of the paper:

  • The first section introduces the problem clearly. I suggest introducing the contribution of the remaining sections of the paper.
  • The references are updated and adequate. Please introduce briefly the obtained results of the main studies in terms of F1-score, sensitivity, and RMSE for each work, if it is possible.
  • To improve the readability of the manuscript, please indicate in a table the number of subjects involved in this study for two datasets.
  • Please indicate the environment in which the experiments were carried out has to be described appropriately. What tool was used for software development?
  • Please improve the Conclusions, this section is too brief.

Author Response

The manuscript describes a denoising system, consisting of a combined hardware and software solution, able to remove the disturbing component typical of acquisitions by dry electrodes and identify fQRS complexes. The performance of the proposed systems was evaluated in terms of sensitivity, F1-score, and root-mean-square error (RMSE) on signals selected from two datasets.

The manuscript is well written and organized. The results seem to be competitive in comparison with existing studies existing in the literature.

We would like to thank the Reviewer for her/his positive feedback.

We have modified the manuscript according to the Reviewer’s observations as explained in more detail in the following. In the revised version, the modified parts are highlighted in red.

Some comments to improve the quality of the paper:

  •     The first section introduces the problem clearly. I suggest introducing the contribution of the remaining sections of the paper.

As suggested by the reviewer, the outline of the manuscript has been added at the end of the introduction.

  •     The references are updated and adequate. Please introduce briefly the obtained results of the main studies in terms of F1-score, sensitivity, and RMSE for each work, if it is possible.

 For the studies that employed the same metrics considered in our study, we reported their results in the Discussion. However, we would like to underline that the proposed method was specifically designed to remove the noise from signals acquired with dry electrodes, while the algorithms presented in the literature were all tested on datasets acquired with traditional wet electrodes. Furthermore, the employed datasets are different for acquisition conditions, electrode position, and instrumentation. Therefore, the comparison provides only an indication of the performance, which was better assessed with the comparison with the VA method, which was done on the same data, both for the acquisitions with dry and wet electrodes.  

  •     To improve the readability of the manuscript, please indicate in a table the number of subjects involved in this study for two datasets.

We added this information in the captions of Tables 2 and 3.

  •       Please indicate the environment in which the experiments were carried out has to be described appropriately. What tool was used for software development?

The acquisitions were carried out at the labour ward of Máxima MC Veldhoven, the Netherlands, and the software was developed in Matlab 2019b (MathWorks Inc., Natick, USA).  Both the statements have been added to the text.

  •   Please improve the Conclusions, this section is too brief.

According to the reviewer’s comment, Section “Conclusion” has been improved.

Round 2

Reviewer 1 Report

Dear Authors,

You have observed the corrections/additions, recommended by the reviewers. Your manuscript can now be considered for publication.